# Jean-Louis Luche and the Interpretation of Sonochemical Reaction Mechanisms

**DOI:** 10.3390/molecules26030755

**Published:** 2021-02-01

**Authors:** Mircea Vinatoru, Timothy J. Mason

**Affiliations:** 1Faculty of Applied Chemistry and Material Science, University “Politehnica” of Bucharest, 1-7 Gh. Polizu, 011061 Bucharest, Romania; 2Faculty of Health and Life Sciences, Coventry University, Priory Street, Coventry CV1 5FB, UK; t.mason@coventry.ac.uk

**Keywords:** Jean-Louis Luche, sonochemistry, electron transfer, ordering effect

## Abstract

Sonochemistry can be broadly defined as the science of chemical and physical transformations produced under the influence of sound. The use of sound energy is rather a young branch of chemistry and does not have the clear definitive rules of other, more established, divisions such as those in cycloaddition reactions or photochemistry. Nevertheless, there are a few guidelines which can help to predict what is going to happen when a reaction mixture is submitted to ultrasonic irradiation. Jean-Louis Luche, formulated some ideas of the mechanistic pathways involved in sonochemistry more than 30 years ago. He introduced the idea of “true” and “false” sonochemical reactions both of which are the result of acoustic cavitation. The difference was that the former involved a free radical component whereas only mechanical effects played a role the latter. The authors of this paper were scientific collaborators and friends of Jean-Louis Luche during those early years and had the chance to discuss and work with him on the mechanisms of sonochemistry. In this paper we will review the original rules (laws) as predicted by Jean-Louis Luche and how they have been further developed and extended in recent years.

## 1. Early Attempts at Understanding Sonochemistry

In the 1980’s while based in Grenoble at Joseph Fourier University Jean Louis Luche began his investigations into the effects of ultrasound on chemical reactions and later started to formulate the rules governing this branch of science [1]. Together with others working in the field it was generally accepted that sonochemistry was the result of acoustic cavitation. This had been clearly identified in a paper published in 1956 entitled “Chemical Effects of Ultrasonics—‘Hot Spot’ Chemistry” [2]. The abstract of this paper claimed that “Experimental evidence is presented which supports the conclusions that chemical processes brought about by ultrasonics require cavitation”. When the general subjects of power ultrasound and chemistry became unified into sonochemistry some of the comments about it suggested that acoustic cavitation might be just a method of achieving efficient mixing. Using ultrasound we could achieve excellent emulsification and dispersion. Acoustic cavitation could also be used for surface cleaning to keep catalysts active or electrochemical reactions more efficient. Such was the case in the early 1990s and many chemists simply viewed the subject as something that was difficult to reproduce and therefore something of a “Black Art” [3]. Many experimentalists found it difficult to reproduce the work of other groups in their own laboratories. These difficulties could be traced to a failure of the original report to specify the exact sonication conditions, e.g., the frequency of ultrasonic irradiation, the precise power entering the reaction system, the geometry of the reaction vessel, the presence of a bubbled gas or even the temperature of the reaction. But these were in a sense “teething troubles” in a subject which has now achieved international acceptance. An experimental protocol was established including even such apparently trivial (but in fact very important) parameters such as the positioning a reaction vessel in a precise position in an ultrasonic cleaning bath. Once the possibilities of different types of ultrasonic systems had been recognized then even in those early years the potential for adoption by industry became apparent [4].

Then attention turned to more fundamental questions about sonochemistry such as what type of reaction was likely to be most affected by sonication and which ones were not. In the beginning for some of us working in the field this question seemed unimportant since it was clear that any reaction involving surface chemistry would almost certainly be enhanced by the application of ultrasound. Most of these could be ascribed to the mechanical effects of cavitation bubble collapse. In 1987, Lorimer and Mason [5] and then Lindley and Mason [6] published consecutive papers in Royal Society of Chemistry Chemical Reviews relating to the physical principles and synthetic aspects of sonochemistry that containing a total of 368 references. Although these articles did not identify specific rules governing sonochemistry an attempt was made to group reactions into four types based on the main interests of a synthetic chemist. The effects of ultrasound were summarized in terms of four different types of reaction.

Reactions involving metal surfaces: divided into those in which the metal is a reagent and is consumed in the process and those in which the metal functions as a catalyst.Reactions involving powders or other particulate matter: the overall reactivity, will depend upon particle size reduction producing greater reactive surface area. and simultaneous surface activation.Emulsion reactions: ultrasound is known to generate extremely fine emulsions with a dramatic increase in the interfacial contact area between the liquids.Homogeneous reactions: in order to explain sonoluminescence or the fragmentation of alkanes must involve more than simply the mechanical effects of cavitation. They may also involve temporary effects on solvent structure resulting in changes to the solvation of reactive species.

Generic classifications such as these helped a chemist to appreciate more fully the results which were emerging involving electrochemistry in plating and synthesis [7,8], electro-analysis [9] and the increasingly important fields of organometallic chemistry [10], phase transfer catalysis [11] together with polymer formation and degradation [12,13]. A recent review on the in situ coupling of ultrasound to electro- and photo-deposition methods identified the latter as a relatively unexplored area of sonochemistry involving free radicals [14]. However, the majority of interest in sonochemistry in the years leading up to 1990 was in organic synthesis [6,15]. 

In 1986, Tim Mason organized the first ever meeting devoted to sonochemistry as part of the Autumn Meeting of the Royal Society of Chemistry held at Warwick University UK. This meeting brought together many of the prominent researchers in the field at that time and it was the first occasion when Tim Mason met Jean-Louis Luche. In the years following that meeting Jean-Louis began to take a particular interest in the underlying rules of sonochemistry. In 1989 he published a review entitled “Sonochemistry—the use of ultrasonic waves in synthetic organic chemistry” which contained 235 references [16]. In the conclusions to that article he wrote about the problems which could delay general applications of this new science and proposed that these would be the result of an incomplete understanding of the underlying theory. Cavitation appeared to explain many things but there were areas that could not be the result of just cavitation. He quoted two examples: one was the almost quantitative yield of adamantanes from the super acid catalyzed rearrangements of some polycyclic hydrocarbons [17] and the other was some work by another pioneer of sonochemistry, Jacques Berlan, involving the reaction of organometallics with a substrate in the solid state [18]. The final paragraph in the Luche review reproduced here in the exact language that he used read:

“To overcome these problems more fundamental research is necessary. In parallel, a satisfactory theoretical understanding will be found more easily by the multiplication of new examples. Thus, the final word of this conclusion that we can address to chemists is both encouraging and short: TRY”.

The first paper which had provided evidence for the specific nature of sonochemistry was published by Ando et al. who described the first case of “sonochemical switching” [19]. The original system which led to this discovery (Scheme 1) consisted of a suspension of benzyl bromide and alumina-supported potassium cyanide in toluene. The stirred reaction provided regioisomeric diphenylmethane products via a Friedel-Crafts reaction between the bromo compound and the solvent, catalyzed by the solid phase reagent. In contrast, sonication of the same constituents furnished only the substitution product, benzyl cyanide. According to the authors, a structural change on the catalytic sites of the solid support could be responsible for this effect. This type of result shows that sonication is definitely not just another method of providing agitation in a medium. This paper provided a key piece of information for those seeking to formulate the rules of sonochemistry.

In 1990, Jean-Louis published consecutive papers in Tetrahedron Letters in which he began to formulate his ideas about mechanisms in sonochemistry [20,21]. Sonochemical reactions were to be divided into three types:Type 1 as a consequence of coordinative unsaturated or free radicals generated by cavitation, in homogenous media—true sonochemistryType 2 caused by only the mechanical effects of sonic waves are operative in heterogenous media—false sonochemistryType 3 “true” sonochemical reactions in which a single electron transfer is involved in a key step.

In the following year, 1991, Mircea Vinatoru visited Tim Mason in Coventry for a one-day course on sonochemistry and following this he went on to visit Jean-Louis Luche who was working in Grenoble at that time. This was his first meeting with Jean-Louis and the three days spent there allowed plenty of time for discussions about sonochemistry. Together they travelled by car to attend the second ESS meeting in Gargnano, on Lake Garda in Italy in September. This was when Mircea became more seriously committed to sonochemistry.

Now the history of the development of sonochemistry progresses to the European Union COST programmes. COST is the acronym for "Cooperation in Science and Technology" an organization founded in 1971 by the European Community to promote inter-European cooperation in any field of scientific and technological research. For many years chemistry had not been a part of COST but the Management Committee of the COST Programme met in Brussels on 9 December 1992 and made the decision to include chemistry for the first time. This was particularly exciting news for Jean-Louis and in a FAX sent to Tim Mason on January 12th, 1993 about this meeting he wrote:

“For the first time, a COST programme is specifically devoted to Chemistry, particularly Chemistry under unusual and extreme conditions. Under this heading, we find high pressure and high temperature chemistries, microwaves, plasma chemistry, reactions in the absence of solvent, and, of course, our common point of interest: Sonochemistry. This is the reason of this letter. During the meeting in Brussels, it appeared clearly that sonochemistry is rather advanced in comparison to other new chemistries. We can envisage the creation of networks involving a multilateral cooperation in various aspects of sonochemistry”.

The overall programme was designated as D6 and within it two networks devoted to sonochemistry were created. Jean-Louis took charge of project number D6/0008/93 “Organic Sonochemistry, Mechanisms and Synthetic Applications” together with G. Descotes Univ. Villeurbanne, S. Toma Univ. Bratislava, R. Miethchen Univ. Rostock, H. Fillion Univ. Lyon, A. Campos Neves Univ. Coimbra and J. Jurczak Academy of Sciences Warszaw. The second network involved Tim Mason in project number D6/0007/93 “Fundamental in Sonochemistry” was led by C. Petrier Univ. de Savoie, and also included J. Reisse Univ. Libre de Bruxelles, T. Lepoint Inst. Meurice Chimie Bruxelles, J. Berlan ENSIGN Toulouse, H. Delmas ENSIGN Toulouse and G. Portenlanger TU München.

In 1996 an important text was produced from that D6 programme entitled “Chemistry Under Extreme or non-Classical Conditions”. It was edited by Rudi van Eldik and Colin D Hubbard and included chapters written by participants in the various sections of COST DOne of the chapters “Ultrasound as a new Tool for Synthetic Chemists” was co-authored by Jean-Louis Luche and Tim Mason [22]. In it the three rules governing sonochemistry were written in the following terms:

Rule 1 applies to homogeneous processes and states that those reactions which are sensitive to the sonochemical effect are those which proceed via radical or radical-ion intermediates. This statement means that sonication is able to effect reactions proceeding through radicals and that ionic reactions are not likely to be modified by such irradiation.

Rule 2 applies to heterogeneous systems where a more complex situation occurs and here reactions proceeding via ionic intermediates can be stimulated by the mechanical effects of cavitational agitation. This has been termed “false sonochemistry” although many industrialists would argue that the term “false” may not be correct, because if the result of ultrasonic irradiation assists a reaction it should still be considered to be assisted by sonication and thus “sonochemical”. In fact, the true test for “false sonochemistry” is that similar results should, in principle, be obtained using an efficient mixing system in place of sonication. Such a comparison is not always possible.

Rule 3 applies to heterogeneous reactions with mixed mechanisms, i.e., radical and ionic. These will have their radical component enhanced by sonication although the general mechanical effect from Rule 2 may still apply. Two situations which may occur in heterogeneous systems involving two mechanistic paths are (a) When the two mechanisms lead to the same product(s), which we will term a “convergent” process, only an overall rate increase results, (b) If the radical and ionic mechanisms lead to different products, then sonochemical switching can take place by enhancing the radical pathway only. In such “divergent” processes, the nature of the reaction products is actually changed by sonication.

Several other examples were summarized nearly 20 years later in a paper by Cravotto and Cintas [23] in which electron transfer promoted by ultrasound was associated with sonochemical reactions. The authors made a comment in the paper which perhaps sums up neatly the state of play at that time:

“If one asks any synthetic sonochemist about the possibility of obtaining distinctive results, other than under conventional conditions, by applying ultrasound to a chemical reaction, he or she will probably give you the same answer: perhaps.”

To this day, apart from the foundations laid down by Jean-Louis Luche as outlined above there have been no further rules brought in that govern sonochemistry to compare with those long-established in photochemistry and thermochemistry. However, the search goes on…

## 2. Explanations for Sonochemical Reactions Based on Electron Transfer

If there are no further rules that can be identified at this stage in the development of sonochemistry then perhaps the way forward is to identify some well documented explanations for sonochemical effects. If these “explanations” are consistent then they might themselves lead on to some new guidelines for sonochemistry. 

Let us return to Ando’s reaction ‘the sonochemical switch’, the authors claimed that potassium cyanide decreases the catalytic activity of alumina [19]. Why is that? No plausible explanation was offered. Using a simple ultrasonic cleaning bath, 45 kHz, 200 W power at 50 °C the authors showed that ultrasonic irradiation completely switched the course of the reaction. They also made the very important observation that if there was no KCN added to the reaction then the Friedel-Crafts reaction occurs under both mechanical agitation and ultrasonic irradiation. It would therefore seem entirely reasonable—as a suggestion—that ultrasound has in some way “assisted the contact of potassium cyanide with alumina to decrease the catalytic ability of alumina”.

However, there is an alternative explanation which is that alumina increases the nucleophilic character of CN^−^. In 1979, Regen and co-workers published a paper entitled “Impregnated cyanide reagents. Convenient synthesis of nitriles” [24]. This was later broadened to other nucleophiles to show that neutral alumina could act as a tri-phase catalyst not only for CN^−^ but also for I^−^, Cl^−^ and acetate in the displacement of bromide from 1-bromooctane [25]. These studies showed that the nucleophile needed to be in intimate physical contact with alumina and that this could be achieved by heating up to 90 °C but there was no ultrasound employed. In the ultrasonic switching experiment however the experiments with and without ultrasound were performed at 50 °C and no nucleophilic substitution was observed in the silent stirred reaction. This indicates that ultrasound must have had a specific effect on CN^−^ on the surface of the alumina. 

The impregnation of alumina with cyanide salt requires only three steps, mixing the aqueous cyanide salt with alumina, filtration and then drying of the cyanide impregnated alumina (4 h, 110 °C, 0.05 mmHg) [24]. However, the drying procedure cannot result in the total removal of water which has a great affinity to alumina. HO groups can still be detected on an alumina surface even after being dried at 1000 °C [26]. Thus, it is reasonable to consider that there will be some water on the alumina surface during this sonochemical switch reaction especially since the alumina was used just as supplied by the manufacturer i.e. not dried. Under these conditions the ionic structure of KCN and the availability of alumina to offer electrons to support KCN ionization may result in the reaction taking place via electron transfer from the cyanide ion to benzyl bromide. Electron transfer reactions are well established in organic chemistry and can be involved in nucleophilic displacement reactions [27]. Single electron transfer (SET) reactions were included in the ideas of Luche who considered that any reactions in which a radical or SET mechanism could be clearly identified should provide useful examples of the efficiency of sonochemical enhancement [22]. 

Among examples of such reactions which are also examples of sonochemical switching is the KornbIum-Russell reaction (Scheme 2) [28].

4-NitrobenzyI bromide reacts with 2-lithio-2-nitropropane via a predominantly polar mechanism to give, as a final product, 4-nitrobenzaldehyde. In contrast, the SET pathway leads to a dinitro compound. Sonication changes the normal course of the reaction and gives preferentially the latter compound, in amounts depending on the irradiation conditions and the acoustic intensity. Under optimal conditions of irradiation, the ratio of C/O alkylation is practically reversed with respect to that of the silent reaction, indicating a direct intervention of sonic waves in the electron transfer process.

A further example of switching is to be found in another paper from the Ando group describing the reaction of lead tetraacetate with styrene [29]. The addition may follow either an ionic or a radical pathway, affording the products depicted in Scheme 3. In the former case, the lead reagent adds to styrene to give a gem-diacetate via a carbocation, whereas the first stage of the radical pathway consists of the decomposition of lead tetraacetate to give the methyl radical, which then reacts with styrene. This was further investigated some years later by a team involving both Ando, Luche and Tim Mason [30].

The idea that sonication promotes electron transfer was certainly part of the mechanistic ideas of Jean-Louis Luche and supported by many others. We propose however that there is a possible nuance to this general classification based on the ideas suggested by Eberson in 1982 that there might be more than one type of electron transfer mechanism in organic chemistry [31].

Much of the original groundwork on electron transfer came from studies of inorganic chemistry and formalized by the 1983 Nobel Prize winner in chemistry Henry Taube. He had a particular interest in the mechanisms of electron-transfer reactions in metal complexes [32]. For the organic chemist the descriptions of electron transfers as applied to inorganic complexes are rather complicated but Eberson simplified the concept by proposing that the idea of electrons being transferred one by one was somewhat at odds with the two-electron centered electronic theory of organic molecules [31]. He described two different mechanisms that he identified as inner and outer sphere:

*Outer Sphere*. This type of electron transfer in organic chemistry involves electron movement through space between molecules which do not associate through any form of partial bond formation (Scheme 4).

*Inner Sphere*. When the two molecules form a temporary bond, effectively a bridge, through which the electron could travel from one molecule to another (as in aromatic nitration (Scheme 5).

Using this concept, it is possible to advance a theory in which sonication might induce either a reaction switch or a reaction acceleration depending on the type of electron transfer involved. This idea was first suggested by Mircea Vinatoru in a paper published in 1994 about the reaction between nitrobenzene with triphenylchloromethane [33]. The reaction led to several products which he suggested arose from a switch of mechanism from outer- to inner-sphere electron transfer. This would involve less energy than electron transfer between discreet molecules—outer sphere electron transfer—which is the more common pathway in organic chemistry.

Now let us re-consider the case of the sonochemical switch from Friedel-Crafts reaction to nucleophilic displacement (Scheme 1) in terms of two types of electron transfer. The “normal” course of the reaction (in the absence of ultrasound) proceeds via alumina interaction with benzyl bromide causing a polarization of the carbon bromine bond then the benzyl positive part attacks toluene (present in excess as the solvent) resulting a mixture of *O*- and *P*-benzyltoluene. However, in the presence of ultrasound the polarized carbon bromine bond undergoes an inner-sphere electron transfer as shown in Scheme 6 followed by radical combination of the resulting benzyl and cyanide radicals.

The authors would like to emphasize that we support the overall idea of “true” sonochemistry as identified by Luche involving electron transfer reactions. We have simply added a second step in the type of electron transfer involved to provide a possible differentiation between electron transfer types as shown in Scheme This was presented by M. Vinatoru in a talk entitled “Exploring Alternative Mechanisms and Pathyways in Sonochemistry” at the 4th Asia-Oceania Sonochemical Society Conference held in Nanjing, China in The scheme suggests a possible use of ultrasound as tool to identify the type of electron transfer involved in a sonochemical reaction depending on whether it involves a switch in products or a rate acceleration.

## 3. Sonochemistry in the Absence of Cavitation

Thus far, it is generally accepted that sonochemistry is the result of the effects of acoustic cavitation. This was first clearly stated in a paper published in 1956 introducing what the authors called “hot-spot chemistry” and in which they wrote that “experimental evidence is presented which supports the conclusions that chemical processes brought about by ultrasonics require cavitation” [2]. However, there are some examples in the literature which are more difficult to rationalize in terms of simple cavitation collapse. 

In May 1994, Tim Mason wrote to Jean-Louis at C.N.R.S. Universite Paul Sabatier “Here in Coventry we will work with Mircea Vinatoru on the ordering effect and electron transfer type reactions. He says he is also working with you. No problems for me but I do not want to be either competing with you or duplicating some studies”. Later, in September that same year Mircea Vinatoru presented a paper at a COST meeting that followed the 4th meeting of the European Society of Sonochemistry (ESS4) held in Blankenberg, Belgium. He discussed a number of unusual observations in sonochemistry including the suggestion that ultrasound can induce order in liquids. At the time this was a new concept for sonochemistry and received a mixed reception although Jean-Louis Luche was among those supporting the ideas. In 1998 Jean-Louis co-authored a paper with Mircea Vinatoru on the sonochemical and thermal redox reactions of triphenylmethane and triphenylmethyl carbinol in nitrobenzene [34]. Although the sonochemical reactions were in low yield the results supported the idea that sonication promotes electron transfer.

The subject of ultrasonic ordering of liquids lay dormant for some years until at the Eleventh Meeting of the European Society of Sonochemistry, held at La Grande-Motte near Montpellier, France in June 2008 Tim Mason gave a plenary address entitled Recent Trends in Sonochemistry and at end of the address presented one slide to remind the audience of the ideas of Vinatoru (Scheme 7).

Some years later in 2014, the ESS 14 in Avignon was dedicated to the memory of Jean-Louis Luche who had passed away on March 24th of that year aged 73 [1]. As outgoing president Tim Mason gave a presentation entitled “Some neglected or rejected paths in sonochemistry—A very personal view” [35]. In the conclusion to this paper he wrote “I would like to think that Jean-Louis Luche would have given serious consideration to all of these neglected or rejected paths and other strange or inexplicable aspects of sonochemistry. Certainly, I do not believe that we as sonochemists, whatever our particular scientific leaning, should discount any new ideas in our subject until they have been thoroughly tried and tested”.

This was the driving force behind the appearance of a new type of paper in Ultrasonics Sonochemistry a so-called discussion paper. In this Vinatoru and Mason hypothesized that sonochemical reactions may occur even in the absence of cavitation [36]. The idea of such a paper is, as its name suggests, to introduce a topic for discussion and further work. In the conclusions the following was written “Under the conditions outlined above the authors believe that it is not always necessary to reach the cavitation threshold to induce sonochemical reactions. In other words, sonochemical reactions could occur in the absence of cavitation. This hypothesis brings with it the need for a new range of experimental sonochemistry targeted at shedding light on the idea of an “ordering effect”. Naturally we would be happy to see some corroboration of our ideas but, as true scientists, we would also be prepared to accept results which disprove it”.

It was therefore a pleasure to find that in 2020 Yeo et al from RMIT University in Melbourne Australia published a review article entitled “High Frequency Sonoprocessing: A New Field of Cavitation-Free Acoustic Materials Synthesis, Processing, and Manipulation” [37]. In this review using much higher frequencies than normally associated with sonochemistry evidence is provided for sonoprocessing in the absence of cavitation.

## 4. Final Comments

Mechanisms in sonochemistry are not yet fully rationalized although the original observations that radical rather than ionic reactions are affected remain valid. Additional ideas have been proposed in terms of different types of electron transfer mechanism and transformations that occur under the influence of ultrasound but without cavitation. 

Overall, we could draw an analogy with photochemistry in that the first law of photochemistry states that light must be absorbed for photochemistry to occur. There is certainly a parallel first law of sonochemistry—sound must be absorbed for sonochemistry to occur.

Maybe—as we enter the realms of hypothesis—there might be two other laws that involve the types of electron transfer induced by ultrasound. A second law defined as ultrasound accelerates outer-sphere electron transfer reactions and even (although more hypothetical at this stage) a third law which suggests that a sonochemical switch in reaction pathways is possible if an inner-sphere electron transfer pathway is possible. Future investigations should be able to prove or disprove these two further laws, but certainly the first law will stay forever.

## Data Availability

Not applicable.

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
