# Peer review of "Jean-Louis Luche and the Interpretation of Sonochemical Reaction Mechanisms"

_molecules, 2021, doi:10.3390/molecules26030755_

Round 1

Reviewer 1 Report

This article represents a fine summary of the work of Jean-Louis Luche with regard to his notions of electron transfer in organic reactions and the possibility of inner- and outer-sphere electron transfer switching, an idea which is used probably more commonly in inorganic and organometallic chemistry than in organic chemistry, but which nonetheless describes some of the reaction pathways observed by Professor Luche. It is a touching summary written by people who knew Jean-Louis well. It will be of interest not only to more senior sonochemists who have been familiar with Jean-Louis's work for a long time but also to the novice and neophyte sonochemist, as it provides mechanistic insights and suggestions for new directions, particularly in organic sonochemistry and hetrogeneous sonochemistry.

Author Response

We wish to thank to the referee for his kind comments.

Reviewer 2 Report

-The authors discussed the sonochemical reaction mechanisms of the interest in organic chemistry, mainly.

-This is important, because in many investigation of this type, the authors are dominantly interested in the final product, but not in mechanism of the corresponding reactions.

- The manuscript is well presented.

Author Response

We would like to thank to the referee for his kind comments.

Reviewer 3 Report

This review-type article by Vinatoru and Mason is related to an interesting and important topic covering the interpretation of sonochemical reactions mechanism, especially focusing on a great scientist contributor in this research area, Prof. Luche.

I’m convinced that this manuscript is a very important contribution in the field of sonochemistry.

I can have just only some minor revisions, they are as follows:

  1. Please check the manuscript for some typos, and rephrase some sentences in the manuscript for a better understanding. Please check in Page 2 : “…fragmentation of liquid alkanes…”
  2. Please incorporate the following scientific work which will certainly complete the well-elaborated bibliographic investigation of the authors: Molecules, 22(2) (2017) 216; doi:10.3390/molecules22020216. 
  3. Please check to include the full address of your references.

Author Response

we want to thank to the referee #3 who has identified some minor revisions that are required and we have answered these as follows:

Adjust sentence

Homogeneous reactions: in order to explain sonoluminescence or the fragmentation of liquid alkanes homogeneous reactions must involve more than simply the mechanical effects of cavitation. They may also involve temporary effects on solvent structure resulting in changes to the solvation of reactive species.

We made the suggested changes. Thank you.

Insert full stop

sonochemistry. In 1989 he published a review entitled “Sonochemistry – the use of ultrasonic …

We made the correction. Thank you

reduce gap by 1 space

then sonochemical switching can take place by enhancing the radical pathway  only.

We made the correction. Thank you.

Add the new reference in this section

Generic classifications such as these helped a chemist to appreciate more fully the results which were emerging involving electrochemistry in plating and synthesis [7, 8], electro-analysis [9] and the increasingly important fields of organometallic chemistry [10], phase transfer catalysis [11] together with polymer formation and degradation [12, 13]. A recent review on the in situ coupling of ultrasound to electro- and photo-deposition methods identified the latter as a relatively unexplored area of sonochemistry involving free radicals [14]. However, the majority of interest in sonochemistry in the years leading up to 1990 was in organic synthesis [6, 15].

We made this suggestion in the manuscript, inserting the reference as is shown above. We thank the referee for this valuable suggestion.

Adjust the following references yellow highlight

  1. M. Vinatoru, T.J. Mason, Can sonochemistry take place in the absence of cavitation? – A complementary view of how ultrasound can interact with materials, Ultrasonics Sonochemistry 52 (2019) 2-5, https://doi.org/10.1016/j.ultsonch.2018.07.036
  2. R. Rezk, H. Ahmed, S. Ramesan, L.Y. Yeo, High Frequency Sonoprocessing: A New Field of Cavitation-Free Acoustic Materials Synthesis, Processing, and Manipulation, Advanced Science, 8 (2021) 2001983, https://doi.org/10.1002/advs.202001983

We mwde the adjustments as suggested. Thank you.